# Mental Strain of Immigrants in the Working Context

**DOI:** 10.3390/ijerph16162875

**Published:** 2019-08-12

**Authors:** Kevin Claassen, Horst Christoph Broding

**Affiliations:** Faculty of Health Department of Human Medicine, Chair of Occupational Medicine and Corporate Health Management, Witten/Herdecke University, Alfred-Herrhausen-Straße 50, 58448 Witten, Germany

**Keywords:** public health, migrant health, epidemiology, occupational medicine, mental health, burnout

## Abstract

Inability to work due to reported mental strain and psychiatric disorders is rising in Germany these days. Meanwhile the country’s net migration is positive. While there is empirical evidence for a healthy migrant effect regarding the physical health in the beginning (mostly accompanied by a subsequent convergence effect), the mental health of migrants remains partly understudied. In order to evaluate the migrant’s share in the rise of reported mental strain in Germany, 4000 employees were surveyed by means of an online access panel. About 16 percent of them revealed a migration background. Their Copenhagen Burnout Inventory (CBI) score is slightly yet significantly above the German autochthonous’ one both using bi- and multivariate analysis, indicating that there is a specific vulnerability rather than a healthy migrant effect regarding mental strain at work.

## 1. Introduction 

Different to the Diagnostic and Statistical Manual of Mental Disorders (DSM) V [1] and the International Classification of Diseases (ICD) 10, the new ICD-11 [2] now contains the burnout syndrome (QD85). The WHO classifies it as an occupational phenomenon under “factors influencing health status”. The symptoms are energy depletion or exhaustion, increased mental distance, negativism or cynicism, as well as reduced efficacy. They have to be work related to justify the diagnosis. The German Version of the ICD-10 [3] contains burnout as a difficulty in coping with life (Z73). 

The direct and indirect costs of psychiatric disorders in Germany, which means those of healthcare, absenteeism and social security benefits, aggregate to 4.8 percent of the Gross Domestic Product (GDP) in 2015, which exceeds the European average of 4.1 percent [4]. It is estimated that in Europe over a third of the population suffers from psychiatric disorders each year [5]. Meanwhile, the Global Burden of Disease Study finds that psychiatric disorders make up for 7.4 percent of the Disability-adjusted Life Years (DALY) worldwide in 2010 [6]. Data concerning the international costs of burnout are still lacking due to the recency of the diagnosis. 

Nevertheless, there is data from German health insurers available. It says that psychiatric disorders are the second most common cause of incapacity for work. Their proportion has risen from 8.9 to 16.5 percent from 2006 until 2017, while they show the highest corresponding duration of the incapacity for work with 38.9 days per case [7,8].

Psychiatric disorders are also the most common cause of early retirement and show a relatively low age of entry with approximately 50 years on average [9]. Women seem to be affected more often or are more sensible recognizing psychiatric disorders [10]. Explicitly with regard to burnout it is stated that the proportion of days of incapacity for work caused by this disease per 1000 employees has risen from 8.1 days in 2004 to 116.7 days in 2017 [11]. 

This shows that occupational psychiatric disorders are a relevant public health issue. There are several attempts to explain it: increased sensitivity and reporting, tertiarization, acceleration and informational overload, increased flexibility demands and the dissolution of boundaries between working and private life [12]. 

Since 2013 and according to the German Occupational Safety and Health Act, business owners have to evaluate and combat certain structural stressors imposed on their personnel [13], (§ 5). Even though this law applies to German businesses, the relevant stressors are similar to many other economies all over the world. 

Regarding the etiology of work related mental disorders there are several theories with one being the stress–strain model by Rohmert and Rutenfranz, which states that stress does not deterministically lead to strain due to the moderation by certain individual and contextual traits [14]. Although there is no linear relation between stress and strain, intensive chronic stress, as specified by the mainly work related Trier Inventory for Chronic Stress (TICS), in fact increases the risk of suffering from depressive symptoms, insomnia and burnout [15]. Thus, models of allostatic load underline the importance of the duration and accumulation of strain as well as habituation to it [16].

Other theories emphasize the ratio between job demand on one side and the margin of control on the other [17], as well as social support [18] and gratification [19], while there are possible satiation and reversing effects of each factor [20]. 

It is criticized that there is no valid and internationally standardized tool to diagnose burnout [21] and the discriminatory power between burnout and differential diagnoses like fatigue and depressive disorders is limited [22]. Nevertheless, there are several scientific measuring concepts with the Maslach Burnout Inventory (MBI) [23] and the Copenhagen Burnout Inventory (CBI) [24] being the most important which correspond in one form or another to the three dimensions mentioned in ICD-11. 

Within the last years the net migration to Germany has been constantly positive [25]. Employees with a migration background are potentially subjected to stressors like language barriers and discrimination against them [26,27]. According to Hofstede countries differ “by dimensions here labeled power distance, uncertainty avoidance, individualism versus collectivism and masculinity versus femininity”, which leads to different organizational structures and potential misfits regarding the individual cultural disposition [28]. Consequently, inter- as well as intrapersonal acculturative stress can arise [29]. Separation from family and friends is in itself a further stressor and implies the discontinuation of social support [30], which is a major protective factor when retained [31]. Immigrants in Germany suffer from social inequalities like an uncertain legal status or a restricted access to mental healthcare [32,33]. Victims of forced displacement are especially vulnerable due to potentially traumatizing experiences. Accordingly, asylum seekers in Germany suffer significantly more often from mental disorders like the Posttraumatic Stress Disorder (PTSD) [34,35]. 

Against that, Krueger and Moriyama [36] were the first to observe the so-called healthy migrant effect which describes the phenomenon of the mean mortality of immigrants being less than the non-migrants’ rate. It is hypothesized that gravely ill migrants simply return to their country of origin while staying part of the databases in the host countries, which is referred to as the salmon bias [37]. Nevertheless, the healthy migrant effect does not seem to rely exclusively on data artifacts [38,39].

There is a lower prevalence of psychiatric disorders in immigrants, when they are compared to their succeeding generations which were born in the host country [40,41,42]. Vega et al. found for example that mood disorders, anxiety and substance abuse among Mexican immigrants in the US were around half as prevalent as among US-born people and US-born Mexicans. This leads the authors to the conclusion that “greater social assimilation increases psychiatric morbidity” [43]. Regarding schizophrenia, which has a strong genetic component, there are several studies contradicting the healthy migrant effect [44,45,46].

While there is further evidence pointing towards a convergence of the migrant’s health and mortality towards the host country’s average in the course of the acculturation process [47,48,49], the migrant’s initially possibly superior health can be explained by selection processes. Those who migrate are the “fittest” in their country of origin [50], because in economic terms their expected wage increase exceeds their expected monetary and psychic costs of migration [51]. Moreover, migrants seem to cope with culturally diverse work teams more efficiently [52]. As a consequence, there is a human capital flight in disfavor of the sending countries [53]. 

This field study aims to evaluate whether for these reasons there is a healthy migrant effect regarding burnout among employees in Germany. 

## 2. Materials and Methods

In 2019, 4000 employees in Germany were asked within the scope of a survey which is based on a job hazard analysis in line with paragraph five of the Occupational Safety and Health Act [13]. For reasons of accessibility and research economics the employees were part of an online access panel. In general, the panel-participants had been recruited actively (telephone and e-mail recruiting) and passively (commercials on websites). For the specific survey there was a random selection within a proportionate stratification representing the sectors and firm sizes in Germany. Selected panel-participants received a personalized invitation link. Unemployed people were excluded. 

The survey focused on the dimension of personal burnout of the CBI, similar to the German version of the Copenhagen Psychosocial Questionnaire (COPSOQ) [54], which is the key dependent variable. The research team explicitly avoided the usage of the term “burnout”. The burnout factor represents the mean of six questions regarding tiredness, physical fatigue, emotional fatigue, the sentiment “I cannot take anymore”, exhaustion, as well as feeling weak and disease-susceptible. The frequency, which is asked for on a scale of five between often and almost never, is matched by the values 100, 75, 50, 25 and 0 (100 = often, 0 = almost never, etc.); representing a vulnerability rather than the actual diagnosis. Missing values of the CBI items were handled using mean imputation, other missing values using listwise deletion.

The authors of the COPSOQ regard ten points as a significant difference. The mean CBI benchmark value of the COPSOQ in Germany is 42, in which manual workers are underrepresented [55].

A migration background was defined as being born outside Germany. This means that someone whose parents migrated to Germany while he or she was an infant is regarded as having no migration background. In order to evaluate the influence of the migration background on the burnout factor migrants were compared to non-migrants using student’s two-sided t-test of mean difference with Welch-approximation due to differing variances. The presence of a time-dependent convergence effect was tested using Pearson’s correlation between the CBI and the duration of residence among the migrants. The potentially important third variables sex, age, general state of health, highest educational attainment, working hours scheme, the count of children below 16 years and other care dependent relatives, wage satisfaction (as an indicator of gratification) and the sense of community at the workplace (as an indicator of social support) were controlled for as non-standardized coefficients within a multiple Ordinary Least Squares (OLS) regression model. Additionally, the respondents were asked about never having experienced discrimination against oneself due to arbitrary attributes on a scale of five between agreeing fully (1) and not agreeing at all (5).

In order to examine the validity of the CBI and its clinical relevance, we asked for chronic psychiatric or behavioral disorders. Targeting respondents with a possibly clinically relevant manifestation, indicated by CBI values greater than or equal to 75, unadjusted odds ratios were calculated. The migration background acts as the exposure or risk factor. 

Beforehand the sample size calculation was done using the software G*Power Version 3.1.9.4 provided by the Heinrich Heine University Düsseldorf. Statistical analysis was carried out with R-Studio version 1.2.1335. The syntax code as well as the data set is available from the authors on reasonable request.

## 3. Results

Among the 4000 respondents, 642 had a migration background as defined by being born outside Germany, which corresponds to 16.05 percent. They had been in Germany for between less than one year and more than 58 years, with 21.23 years on average.

The studied employees with and without a migration background do not show any striking heterogeneity in the most important demographic third variables as presented in Table 1. Eleven respondents did not tell whether they have a migration background, which results in a net case number of 3989.

Calculating the CBI, in 113 cases mean values were imputed for at least one of the six precursive items. Nonresponse on the CBI items was with 2.99 percent in respondents without a migration background and 2.02 percent within the German group, approximately equally distributed. 

The mean CBI score in the sample is 35.78. It is 35.11 ± 22.47 for employees without a migration background. The burnout inventory value of the employees with a migration background is 38.96 ± 21.87. The resulting t-distributed mean difference of 3.86 is significant (*p* < 0.01). The 95 percent confidence interval puts out a mean difference between 1.97 and 5.75 towards less mental strain for the German autochthones. The comparison is visible between the boxplots in Figure 1.

Within the multivariate OLS regression analysis of the CBI, the migration coefficient reduces to 3.01 compared to the mean difference (or a possible bivariate regression model). An illustration of the values of the other regression coefficients is found in Table 2. The model’s statistical goodness can be assessed by looking at the coefficient of determination. The variance of the independent variables presented in Table 2 explains a quarter of the variance of the CBI values (R² = 0.25). The inclusion of these variables seems to be reasonable looking at the minimally smaller adjusted coefficient of determination (adj. R² = 0.24). 

The duration of the migrant’s residence in Germany and their CBI score are uncorrelated (*r* = 0.03). Hence, there is no convergence effect on the mean level observable. Given that migrants arrive in cohorts, the correlation can still be suppressed by socio-demographic confounders. 

The mean experiences with discrimination against oneself hardly differ between those with and without a migration background (1.87 vs. 1.84). 

There is a small to moderate correlation between the CBI and chronic psychiatric or behavioral disorders (*r* = 0.24). The five percent of the respondents who suffer from the latter show a mean CBI of 58.31. There is no significant difference between migrants (4% affected) and non-migrants (5% affected).

Seven percent of the respondents show CBI values above 74. The unadjusted odds ratio for this manifestation as disease and the migration background as exposure is 1.13 (*n* = 3989). The 95 percent confidence interval includes values between 0.83 and 1.55 indicating that there is, despite a slight tendency, no significantly higher risk for clinical burnout in working migrants in Germany on the mean level.

## 4. Discussion

Though existent, the obtained difference between employees in Germany with and without a migration background is not to be regarded as high. It is not attributable to discrimination as well, although the experience of discrimination in both groups appears to be rare overall, which could suppress intergroup differences. The rise of reported work related mental strain, burnout and other psychiatric disorders is an issue that affects employees in Germany independently from their origin. For the eight predominantly significant explanatory variables, the small to moderate coefficient of determination regarding the multivariate OLS regression model testifies to the multicausality and multidimensionality of the phenomenon of mental strain that still has to be illuminated further and on a highly detailed level. It is therefore advisable that future research focuses on the different national as well as cultural origins.

Both group’s CBI score is less than the German COPSOQ mean of 42 which suggests that those employees having the time and energy to participate in an online survey (be it during work or free time) are less mentally strained than the average German population. It is possible that the sampled mean CBI would increase if the data collection method was changed.

In particular, the migrant’s data could be biased to the extent that the sampled population was derived from an online panel and that on average they have already been in Germany for a relatively long period. This means that the migrants of the sample frame might differ from the migrants of the overall target population of employees in Germany, for example with regard to language skills and socio-economic status. As a consequence, the mental health of the sampled migrants could be overvalued due to the nexus between health and socio-economic status [56]. 

Nevertheless, the percentage of migrants within the sample (16.05%) corresponds to the percentage, of Germans and foreigners with their own migration experiences, in Germany as a whole in 2017 (16.16%) [57]. 

Although less frequently than the longer existing MBI, the translated CBI could be validated cross-culturally [58] and has already been used for comparisons between native and foreign employees [59]. The higher CBI mean value of the employees affected by psychiatric disorders is indicative of the internal validity of the instrument.

A possible limitation is that the COPSOQ version of the CBI only focuses on the dimension of personal burnout, which might disregard the two other ICD-11 symptoms of distance, negativity, cynicism on the one hand and reduced work efficacy on the other. Furthermore, the differentiation from not work related, personal and societal factors proves to be difficult [60].

In 113 cases, the mean had to be imputed due to missing values on one of the variables that constitute the COPSOQ CBI (tiredness, physical fatigue, emotional fatigue, the sentiment “I cannot take anymore”, exhaustion, feeling weak and disease-susceptible). Because nonresponse and migration status are uncorrelated, there is no bias of the analysis of the primary research question. 

Besides that, the data does not enable a precise distinction between voluntary and forced migration. The majority of the more than one million asylum seekers, who have applied since 2015 in Germany [61], is probably not yet part of the online access panel due to language deficits and diverging priorities like finding a job. Thus, it is possible that the sampled migrant’s mental health overestimates the mental health of the population for another reason. It is therefore desirable that future research covers respondents with and without migration and flight background who are not acquirable online in order to differentiate on a more detailed level.

Following the obtained results there is no need for a political health intervention or a migrant group specific burnout prevention, for example on the part of German health insurers. That does not mean that more prevalent stressors, for instance within the realms of ergonomics or dangerous materials, can be ruled out in general. An analysis of the distribution of precarious employment and multi-jobbing is also desirable for future research. 

## 5. Conclusions

This is one of the first cross-sector studies on burnout in migrants in Germany. A mentally healthy migrant effect of employees could not be observed. Instead of a protecting influence, their migration background appears to be a risk factor for mental strain and burnout. Due to the minor effect size, the results do not back up a requirement for migrant specific prevention programs. Nevertheless, work related mental strain and the burnout syndrome are relevant public health issues for both employees with and without a migration background. 

## Figures and Tables

**Figure 1 ijerph-16-02875-f001:**
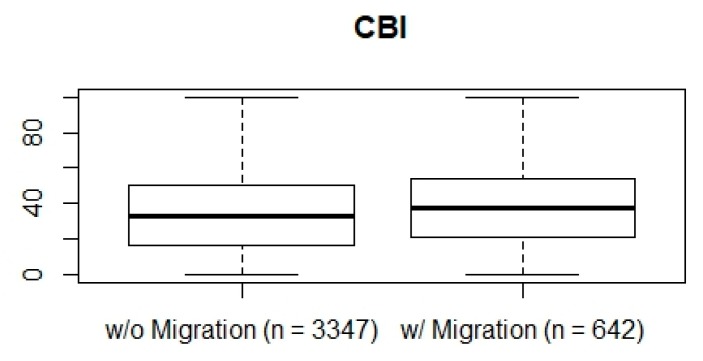
Boxplot of the Copenhagen Burnout Inventory (CBI) values of employees without (*n* = 3347) vs. with a migration background (*n* = 642).

**Table 1 ijerph-16-02875-t001:** Descriptive statistics of the employees without (*n* = 3347) and with a migration background (*n* = 642).

Variable	Without Migration Background	With Migration Background
*n*=	3347	642
Mean Age in Years	41.90 ± 11.60	39.38 ± 11.40
Men %	55	48
Full Time %	87	84
University Degree %	27	29
Satisfied with One’s General Health %	55	56
Satisfied with One’s Wage %	47	48
Satisfied with Sense of Community at Work %	51	51
Mean Count of Children below 16 Years	0.48 ± 0.78	0.46 ± 0.83
Taking Care of Other Relatives %	0.10	0.11

**Table 2 ijerph-16-02875-t002:** Coefficients of the regression of the CBI on migration background and third variables (*n* = 3858).

Variable	Estimate	Significance
*n*=	3858	*p*=
Intercept	15.06	<0.001
Migration Background (binary)	3.01	<0.001
Part Time Work (binary)	0.51	0.542
Male Sex (binary)	–9.67	<0.001
Age	–0.1	<0.001
Sense of Community at Work(decreasing from 1 to 5)	1.94	<0.001
Educational attainment	0.16	0.468
General Health(decreasing from 1 to 5)	6.53	<0.001
Wage Satisfaction (decreasing from 1 to 5)	3.08	<0.001
Care-dependent Relatives (binary)	4.75	<0.001
Children below 16 Years	1.22	0.002

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
