# Peer review of "Mental Strain of Immigrants in the Working Context"

_ijerph, 2019, doi:10.3390/ijerph16162875_

Round 1

Reviewer 1 Report

Thank you for the paper that was aimed to evaluate if there is burnout among migrant and non-migrant employees in Germany. However, there are some methodological issues which includes presentation of results. Please find my comments below:

Introduction

Line 36: “…absence…” of???

Minor editing such as Line 72-73; I think it is one paragraph

The essence of the study in the context of Germany needs to be clear.

Methods

Line 108--The authors need to justify why their population of study were employees who were part of an online access panel managed by prolytics market research.

Line 110 what is this sign §?

Line 122: What if someone was born outside Germany but the parents migrated while he/she was still a baby? How did you handle that? The inclusion and exclusion criteria needs to be clear.

Line 125-127: How does Pearson’s correlation tests moderating effect if no correlation?

Results

Line 164-165: Reduces? The coefficient 3.01 is positive. Those with migration background have a higher mean score than those without.

Line 155-163: Looking ate the confidence interval in the box plot it does not indicate a significant difference between employees without with migration background.

In Line 134-136 you mentioned the calculation of odds ratios. What did you use, logistic regression? If yes, why is this not mentioned in the methods?

Line 177-180: Is this adjusted odds ratio? Why is it contradictory of the finding of OLS which is statistically significant? There is confusion in you results: box plot not significant, OLS significant, odds ratio not significant.

Discussion

This will change when the above issues are addressed.

What is the policy implication of your finding?

What are the potential biases associated with selecting a sample from employees who were part of an online access panel managed by prolytics market research?

Reviewer 2 Report

My major concern with this study is that immigrants are treated as a group with no sense of relevance to their country of origin, culture, and belief system, which indeed has a strong impact on their sense of wellbeing spiritually and health wise. So I recommend that the authors incorporate that data, because this study is very relevant to the field and needs to be published and create yet a new area of research. PTSD and immigration is a growing field of interest and to pursue it to the highest standards is key.

Reviewer 3 Report

This manuscript describes a study designed to test the association of migration status on work-related mental strain and burnout, using a sample of 4000 online respondents living in Germany.  The authors found a small, but statistically significant increase in mental strain/burnout among migrants vs. native-born Germans as measured by the CBI.  The authors concluded that while migration is a risk factor for work-related mental strain, the increased risk does not warrant intervention programs for migrants in Germany.

This paper has some strengths, including addressing a potentially significant public health issue of mental strain among migrant workers and a large sample size with power to detect differences.  However, the limitations seriously detract from the strengths and weaken the conclusions to be made from the results. 

1.       The sample section effect is clearly an issue – and, as the authors themselves note, may have resulted in diluted effect of mental strain, both migrant and non-migrant groups scoring below the average of 42.  Low mental strain scores undermine the assertion that this is a public health issue which needs to be addressed. 

2.       There should be sensitivity analyses to compare the online sample with the general working population in Germany.  Further, there is no information on the non-respondents, e.g., how many there were if any, and their demographic and other characteristics.  Therefore, the external validity of the results, given the sample, remains in question.

3.       The statistical difference found between the two groups may not be clinically significant.  The authors mention useful outcomes like days of work missed over a given period, or psychiatric illness, but these indices were not assessed in relation to the sample CBI scores.  There is no way to assess how the CBI relates to these specific outcomes of interest – which would be the most relevant to public health.

4.       It is unclear whether regression coefficients were standardized or not. Standardization would allow for direct comparison to each other in terms of the relative contribution to the CBI score variance. 

5.       Authors mention discrimination not being a factor at the beginning of the discussion section, but the discrimination item/score referenced in the “Methods” section is not addressed in the results or shown in Tables 1 or 2.

6.       As noted in the limitations, the authors were unable to differentiate between voluntary migration vs non-voluntary (e.g., refugee, asylum-seekers, etc.) migration.  This likely had implications for the CBI scores.

7.       In the “Materials and Methods” section, the authors mention a group of respondents with CBI scores of 75 or higher – stating that ORs were calculated, but the migration effect among the 7% who fell into this group is non-significant. How does this square off with the larger sample, as well as the general working population?

8.       The manuscript needs to be edited and written up to standards.  There were numerous instances of awkward phrases (e.g. p. 7, “Therefore, especially the distribution of precarious employment and multi-jobbing had to be analyzed, which is also desirable for future research” ), jargon words (e.g., “ex ante,” “autochthones,” and “noxae”), and structural issues with paragraph alignment, etc., making the paper difficult to read at times.

Reviewer 4 Report

Lines 29-31: Different to the Diagnostic and Statistical Manual of Mental Disorders (DSM) V [1] and the 30 International Classification of Diseases (ICD) 10, the new ICD-11 [2] will contain a code for the 31 burnout syndrome (QD85) subsumed under “factors influencing health status”

This may be rephrased as “Burnout syndrome is now included in ICD11, with the WHO validating it as an “occupational phenomenon” under “factors influencing health status” or something along those lines.

Line 60-61: the stress-strain-model by Rohmert and Rutenfranz, which states that stress does not deterministically lead to strain due to the moderation by certain individual and contextual traits. This reference is from 1975 and in German, without any translation available (Rohmert, W.; Rutenfranz, J. Arbeitswissenschaftliche Beurteilung der Belastung und Beanspruchung an 265 unterschiedlichen Industriearbeitsplätzen; Bonn, 1975).

Line 62-64 Although there is no linear relation between stress and strain, intensive chronic stress, as specified by the mainly work related Trier Inventory for Chronic Stress (TICS), in fact increases the risk of suffering from depressive symptoms, insomnia and burnout – the reference listed for this #15 does not address this (possibly because only abstract is available in English).  

Another paper I found, titled “Ergonomics: concept of work, stress and strain” by Rohmert (entire article in English) helped me understand the concept of stress and strain.   https://onlinelibrary.wiley.com/doi/epdf/10.1111/j.1464-0597.1986.tb00911.x

As with many of the other references, was unable to find the English translations of at least the abstract (which makes it difficult to fully evaluate the paper).

The references do not include other studies in which the Copenhagen burnout inventory has been used in eliciting differences between native and immigrant workers. The following is one such reference which points out many interesting factors: Lin LP, Wu TY, Lin JD. Comparison of job burnout and life satisfaction between native and foreign female direct care workers in disability institutions. Work. 2015;52(4):803-9.

Line 182-183: The obtained difference between employees in Germany with and without migration background is though existent not to be regarded as high.

The study does not lend itself to the conclusions drawn, especially given what is pointed out by the authors (lines 191-198):  “Both group’s CBI score deceeds the German COPSOQ mean of 42 which suggests that those employees having the time and energy to participate in an online survey (be it during work or free time) are less mentally strained than the German average population. Especially the migrant’s data could be biased to the extent that the sampled population was derived from an online panel and that they are on average already for a relatively long period in Germany. This means that the migrants of the sample frame might differ from the migrants of the overall target population of employees in Germany for example with regard to language skills and socio-economic status.”  

Reviewer 5 Report

Overall, this is a worthwhile study exploiting a useful dataset on work-related stress of migrants versus non-migrants.

Introduction

While the introduction provides relevant evidence to underline why work-related stress is a significant public health issue, it could be more focused, e.g. by not limiting the discussion of the cost of psychiatric disorders in general (as opposed to specifically work-related stress) and its effect on GDP/DALY/incapacity for work.

Measure of over-qualification or occupational class – without accounting for the fact that many migrants are overqualified for the positions they work in you would not realistically be able to observe a net healthy migrant effect. Without such controls in the model the usefulness of the results is limited.

The comparison of (first generation ie foreign-born) immigrants’ mental health to that of second generation immigrants is not relevant for this paper as the comparison is with native born (line 92) (ignoring that of course the second generation are part of the native born comparison group).

The discussion on why there should be a healthy migrant effect in the first place (rather than reasons why migrants could be at a disadvantage in health or why there is a convergence to native health levels could be extended, given that this is the focus of the paper.

Materials (data) and methods

Given that this is not a probability sample of the general working population, it would be useful to know more about the online access panel – how it is recruited, what occupational composition does it have and in how far is it a fair representation of the general working population (e.g. % migrants in sample vs population).

Information on whether the COPSOQ has been validated for cross-cultural use is missing.

Regarding the odds ratios (line 135) I am not clear whether they are calculated using a model with the same controls as explained for the OLS model.

Results

Table 2 needs standard errors or exact p-values.

Results from line 173 – 180: I can’t follow this well as this is not described in detail. If I understand this right it is the Pearson’s correlation, i.e. bivariate? Given that migrants will be heterogeneous across arrival cohorts (e.g. with respect to their education and hence occupation) I can’t see how a binary test could be a meaningful test of a convergence effect, this must be tested in a multivariate model

Details

Line 72, 73 should have no page break

Line 93 – foreign-born succeeding generations – I don’t understand this term, does this mean second generation migrants? So there are not foreign-born?

Notes on English terms – line 183 – it should be “discrimination” not “discrimination against”. Various places: one can’t translate “bisherige Dauer” as “so far duration of residence”. The literature generally uses the term “duration of residence” and it is implicit that this is only the duration up to the point of measurement.

Line 135 past migration process – Does this mean migrant background ie binary variable or duration of residence ie continuous?

Round 2

Reviewer 1 Report

There is a little improvement. No comment.

Reviewer 3 Report

Authors responded adequately to concerns raised.  

Reviewer 5 Report

Thank you for your responses to the comments I made.

Odds ratios: I now understand that the odds ratios are unadjusted, though I don’t understand why one would report these unadjusted odds ratios at the very end of the results, after discussing a multivariate model. I understand that one measure is about risk of disease, while the other is a continuous score, however, an unadjusted odds ratio could surely only be the starting point for an analysis? For a reader who does not follow the paper very closely, the end of the results section (“The 95 percent confidence interval includes values between 0.83 and 1.55 indicating that there is despite a slight tendency no significantly higher risk for clinical burnout in working migrants in Germany”) could easily be mis-understood as the result of a multivariate analysis (because typically, the latter parts of a results section reports multivariate results). Given that, I suggest making it explicit that this is unadjusted.

Convergence effect: In a cross-sectional analysis where duration of residence is collinear with arrival cohorts one cannot use a bivariate test to determine whether there is a convergence effect of migrants’ CBI scores. Using such a test assumes that all potential confounders (such as level of education, occupational class, age) are distributed similarly across migrants of differing residence durations (which is almost certainly not the case, especially for age). In this situation, all we know from a bivariate test is that CBI scores don’t differ across migrants with duration of residence, but this could be entirely due to compositional differences across cohorts. So results should make this clear.

Overqualification: While I don’t know the specific situation in Germany, it is well established for many developed countries that migrant workers are more often overqualified (overeducated) than native-born workers (see e.g. overview in Béla Galgóczi, Janine Leschke (editors): EU Labour Migration in Troubled Times: Skills Mismatch, Return and Policy Responses. Routledge 2012; pp. 84). Chiswick (1978) and Chiswick and Miller (2007) explain why immigrants would be more often overqualified than native-born workers due to imperfect transferability of human capital.

Note on methods section: The paper would be easier to follow if the description of methods would be in the same order as they are presented in the results.

Data: There are potentially large sample selection effects, both within migrants (as the authors point out themselves in line 211-216) and non-migrants, in the data.  It is very unfortunate that the data has no occupational measure as a) occupational group plays an important role in explaining job demands and stressors, and the prevalence of burn out differs across occupational groups and b) the occupational composition almost certainly differs between migrants and native borns in the sample (or at least, it does so in the working population overall, and it should in the sample if this was representative of the general working population).

The combination of not being able to ascertain the relevant characteristics of the sample and not being able to control for occupational group in the analysis severely limits the insights one can gain from the analysis.
